# Controllable Chest X-Ray Report Generation from Longitudinal Representations

**Francesco Dalla Serra**[1,2]   **Chaoyang Wang**[1]

**Fani Deligianni**[2]   **Jeffrey Dalton**[2]   **Alison Q O'Neil**[1,3]

[1]Canon Medical Research Europe, Edinburgh, United Kingdom
[2]University of Glasgow, Glasgow, United Kingdom
[3]University of Edinburgh, Edinburgh, United Kingdom
`francesco.dallaserra@mre.medical.canon`

## Abstract

Radiology reports are detailed text descriptions of the content of medical scans. Each report describes the presence/absence and location of relevant clinical findings, commonly including comparison with prior exams of the same patient to describe how they evolved. Radiology reporting is a time-consuming process, and scan results are often subject to delays. One strategy to speed up reporting is to integrate automated reporting systems, however clinical deployment requires high accuracy and interpretability. Previous approaches to automated radiology reporting generally do not provide the prior study as input, precluding comparison which is required for clinical accuracy in some types of scans, and offer only unreliable methods of interpretability. Therefore, leveraging an existing visual input format of anatomical tokens, we introduce two novel aspects: (1) *longitudinal representation learning* – we input the prior scan as an additional input, proposing a method to align, concatenate and fuse the current and prior visual information into a joint longitudinal representation which can be provided to the multimodal report generation model; (2) *sentence-anatomy dropout* – a training strategy for controllability in which the report generator model is trained to predict only sentences from the original report which correspond to the subset of anatomical regions given as input. We show through in-depth experiments on the MIMIC-CXR dataset (Johnson et al., 2019a,b; Goldberger et al., 2000) how the proposed approach achieves state-of-the-art results while enabling anatomy-wise controllable report generation.

## 1  Introduction

A chest X-Ray (CXR) is a frequently performed radiology exam (NHS England and NHS improve-

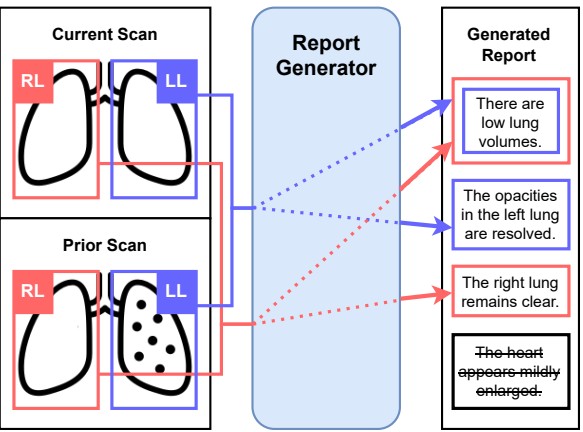

Figure 1: Illustration of our controllable automated reporting system using longitudinal representations. The report generator is trained to generate only sentences corresponding to the selected input anatomical regions. LL indicates the left lung and RL the right lung and we colour-match each region with the corresponding sentence. ~~Strikethrough text~~ represents the section of the report that we do not want the generated report to include when only the LL and RL are selected as inputs.

ment, 2022), used to visualise and evaluate the lungs, the heart, and the chest wall. Given a CXR, a radiologist or a trained radiographer will diagnose disease (*e.g.* lung cancer, scoliosis), and assess the position of treatment devices (*e.g.* tracheostomy tubes, pacemakers). They record their findings in a radiology report, writing a detailed text description of the presence/absence and location of relevant clinical findings. When a prior image is available, the radiologist commonly compares the current clinical findings of a patient with the prior clinical findings to assess their evolution over time (*e.g.* "the heart remains enlarged", "the catheter has been removed"); this is especially critical in follow-up exams performed for monitoring.

Radiology reporting is a time-consuming pro-

cess, and scan results are often subject to delays. In many countries, this reporting backlog is only likely to worsen due to increasing demand for imaging studies as the population ages, and to the shortage of radiologists (Rimmer, 2017; Cao et al., 2023). One strategy to speed up reporting is to integrate automated reporting systems. However, to be employed in real-world clinical scenarios, an automated system must be accurate, controllable and explainable; these criteria are difficult to meet on a task requiring sophisticated clinical reasoning across multiple input image features, and targeting an ill-defined and complex text output.

Previous works on CXR automated reporting have mostly focused on solutions to improve clinical accuracy, e.g. Miura et al. (2021). However, they generally use a single radiology study as input to generate the full report, precluding comparison with prior scans. They also do not allow the end user control over what parts of the image are reported on, leading to limited transparency on which image features prompted a specific clinical finding description; interpretability is currently achieved by generating heatmaps that are often dubious (Chen et al., 2020, 2021). In this work, we hence focus on two novel aspects: (1) **longitudinal representations** – the most recent previous CXR from the same patient is passed as an additional anatomically aligned input to the model to allow effective comparison of current and prior scans; (2) **controllable reporting** – to encourage the language model to describe only the subset of anatomical regions presented as input: this might be single anatomical regions (*e.g.* {cardiac silhouette} → "the cardiac silhouette is enlarged"), multiple anatomical regions (*e.g.* {left lung, right lung} → "low lung volumes") or the full set of anatomical regions (in which case the target regresses to the full report, as in previous methods). A high-level representation is shown in Figure 1.

To summarise, our contributions are to:

1. propose a novel method to create a longitudinal representation by aligning and concatenating representations for equivalent anatomical regions in prior and current CXRs and projecting them into a joint representation;

2. propose a novel training strategy, *sentence-anatomy dropout*, in which the model is trained to predict a *partial* report based on a sampled subset of anatomical regions, thus

training the model to associate input anatomical regions with the corresponding output sentences, giving controllability over which anatomical regions are reported on;

3. empirically demonstrate state-of-the-art performance of the proposed method on both full and partial report generation via extensive experiments on the MIMIC-CXR dataset (Johnson et al., 2019a,b; Goldberger et al., 2000).

## 2 Related Works

### 2.1 Automated Reporting

The task of generating a textual description from an image is generally referred to as image captioning. Advancements in the general domain have often inspired radiological reporting approaches (Anderson et al., 2018; Cornia et al., 2020; Li et al., 2020; Zhang et al., 2021). However, the two tasks differ; the target of image captioning is usually a short description of the main objects appearing in a natural image, whereas the target of radiological reporting is a detailed text description referring to often subtle features in the medical image. Research on CXR automated reporting has focused on improving the clinical accuracy of the generated reports by proposing novel model architectures (Chen et al., 2020, 2021), integrating reinforcement learning to reward factually complete and consistent radiology reports (Miura et al., 2021; Qin and Song, 2022), grounding the report generation process with structured data (Liu et al., 2021; Yang et al., 2022; Dalla Serra et al., 2022), and by replacing global image-level features extracted from convolutional neural networks (He et al., 2016; Huang et al., 2017) as the input visual representations with anatomical tokens (Dalla Serra et al., 2023).

### 2.2 Longitudinal CXR Representation

The problem of tracking how a patient's clinical findings evolve over time in CXRs has received limited attention either generally or for the application of CXR reporting, despite this being a critical component of a CXR report. Ramesh et al. (2022) avoid the problem by proposing a method to remove comparison references to priors from the ground truth radiology reports to alleviate hallucinations about unobserved priors when training a language model for the downstream report generation task. Bannur et al. (2023) introduce a self-supervised multimodal approach that models longitudinal CXRs from image-level features

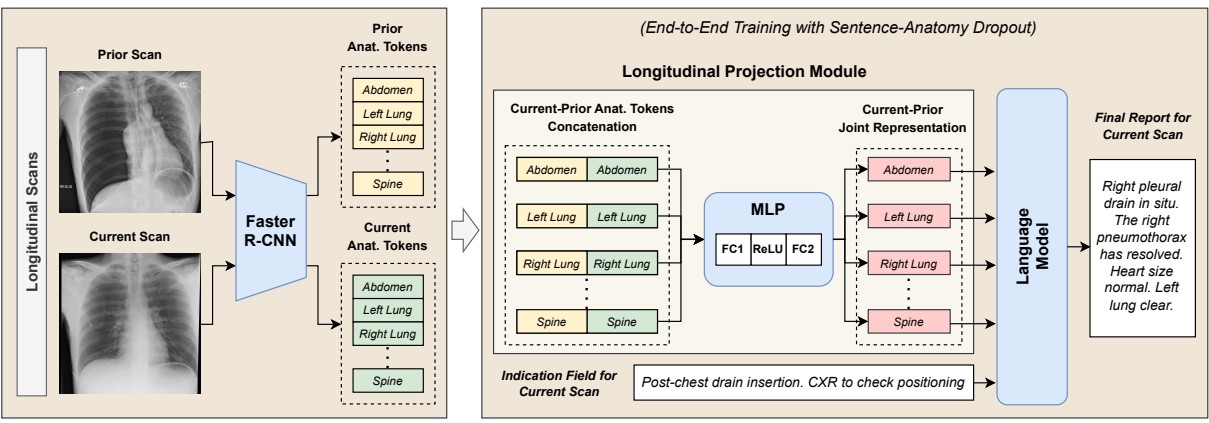

Figure 2: Architecture overview. The anatomical region representations of the current and prior CXRs are extracted from Faster R-CNN (*visual anatomical token extraction*). These are aligned, concatenated and projected into a joint representation (*longitudinal projection module*), then passed alongside the tokenised indication field as input to the language model to generate the report for the current scan. The Report Generator is trained end-to-end using *sentence-anatomy dropout*.

as a joint temporal representation to better align text and image. Karwande et al. (2022) have proposed an anatomy-aware approach to classifying if a finding has improved or worsened by modelling longitudinal representations between CXRs with graph attention networks (Veličković et al., 2018). Similar to Karwande et al. (2022), we project longitudinal studies into a joint representation based on anatomical representations rather than image-level features. However, we extract the anatomical representations from Faster R-CNN (Ren et al., 2015), as in Dalla Serra et al. (2023).

### 2.3 Controllable Automated Reporting

We define a *controllable* automated reporting system as one which allows the end users to select what regions in the image they want to report on, giving a level of interpretability. This has partially been tackled using hierarchical approaches (Liu et al., 2019), by introducing a multi-head transformer with each head assigned to a specific anatomical region and generating sentences exclusively for that region (Wang et al., 2022), and contemporaneously to our work by Tanida et al. (2023) who (similarly to us) generate sentences based on region-level features extracted through Faster R-CNN (Ren et al., 2015).

Tanida et al. (2023) make the assumption that each sentence in the report describes at most one anatomical region. Conversely, we acknowledge that there may be multiple anatomical regions which are relevant to the target text output, *e.g.*,

a sentence "No evidence of emphysema." requires information from both left and right lungs; this requires us to identify valid subsets of anatomical regions in each CXR report for our dropout training strategy (Section 3.4).

## 3 Method

We first extract anatomical visual feature representations $\vec{v} \in \mathbb{R}^d$ with $d$ dimensions for $N$ predefined anatomical regions $A = \{a_n\}_{n=1}^N$ appearing in a CXR. To model longitudinal relations, we perform feature extraction for both the scan under consideration and for the most recent prior scan, and then combine region-wise using our proposed longitudinal projection module. We then input the features to the language model (LM) alongside the text indication field, which is trained using our anatomy-sentence dropout strategy. We show the proposed architecture in Figure 2 and describe these steps in more detail below.

### 3.1 Visual Anatomical Token Extraction

For the anatomical representations, we extract the bounding box feature vectors from the Region of Interest (RoI) pooling layer of a trained Faster R-CNN model. Faster R-CNN is trained on the tasks of *anatomical region localisation* in which the bounding box coordinates of the $N$=36 anatomical regions (*e.g.* abdomen, aortic arch, cardiac silhouette) are detected in each CXR image, and *finding detection* in which presence/absence is predicted in each proposed bounding box region

| Original Report | Anatomical region | Target output | Mapping type |
|---|---|---|---|
| "The mediastinum is mildly enlarged. Blunting of right costophrenic angle noted. No suspicious nodules seen. No pneumothorax or infective consolidation. Bilateral atelectasis, likely post-operative. Degenerative changes seen in both shoulders. NG tube tip positioned correctly in stomach. No free air under diaphragm" | `mediastinum` | "The mediastinum is mildly enlarged." | one-to-one |
| | `abdomen` | "NG tube tip positioned correctly in stomach. No free air under diaphragm." | one-to-many |
| | `left clavicle, right clavicle` | "Degenerative changes seen in both shoulders." | many-to-one |
| | `right lung, left lung` | "Blunting of right costophrenic angle noted. No suspicious nodules seen. No pneumothorax or infective consolidation. Bilateral atelectasis, likely post-operative." | many-to-many |

Table 1: Example correspondences between anatomical regions and target output, for a synthesised CXR "Original Report" (Findings section). Note the different types of correspondence mapping.

for a set of 71 predefined findings (*e.g.* `pleural effusion, lung cancer, scoliosis`).[1] Specifically, we augment the standard Faster R-CNN architecture head — comprising an anatomy classification head and a bounding box regression head — with a multi-label classification head, following Dalla Serra et al. (2023), to extract *finding-aware anatomical tokens* $V = \{\vec{v}_n\}_{n=1}^N$ with $\vec{v}_n \in \mathbb{R}^d$ where $d$=1024. Then, for each anatomical region we select the bounding box representation proposal with the highest confidence score. When the anatomical region is not detected, we assign a d-dimensional vector of zeros. For more details about the model architecture, the loss term and other implementation details, we refer to Dalla Serra et al. (2023).

### 3.2 Longitudinal Projection Module

Taking the *current scan* (the most recent scan at a specific time point) and the CXR from the most recent study (*prior scan*)[2], we extract from Faster R-CNN the anatomical tokens of both CXRs. There is one token for each of the $N$ anatomical regions: $V_{current} = \{\vec{v}_{c,n}\}_{n=1}^N$ and $V_{prior} = \{\vec{v}_{p,n}\}_{n=1}^N$. When the current scan is part of an initial exam: $\vec{v}_{p,n} := \vec{0} \in \mathbb{R}^d \; \forall n = 1, \ldots, N$. We select indices for the subset of regions that we want to report on

$A_{target} \subseteq A$, and we obtain the longitudinal representation by concatenating the anatomical tokens for each anatomical region $a_n$ of the two CXRs, and passing them through the longitudinal projection module $f$:

$$\vec{v}_{joint,n} = \begin{cases} f([\vec{v}_{c,n}, \vec{v}_{p,n}]) & \text{if } a_n \in A_{target} \\ f([\vec{0}, \vec{0}]), & \text{otherwise.} \end{cases}$$

The projection layer $f$ is a Multi-Layer Perceptron (MLP) consisting of a stack of a Fully-Connected layer (FC1), a Batch Normalization layer (BN) and another Fully-Connected layer (FC2). We refer to the resulting output as the current-prior joint representation $V_{joint} = \{\vec{v}_{joint,n}\}_{n=1}^N$.

### 3.3 Language Model (LM)

This consists of a multimodal Transformer encoder-decoder, which takes the current-prior joint representation $V_{joint}$ and the indication field[3] $I$ as the visual and textual input respectively, to generate the output report $Y$:

$$Y = \text{LM}(V_{joint}, I)$$

where $Y$ corresponds to the partial report if $A_{target} \subset A$ or the full report if $A_{target} = A$.

Similarly to Devlin et al. (2019), the input to the LM corresponds to the sum of the textual and visual

---

[1]Full lists of anatomical regions and clinical findings are provided in Appendix A.

[2]A prior scan is only available if the current scan is not part of an initial exam.

[3]The indication field contains relevant medical history in the form of free text and it is available at imaging time.

*token embeddings*, the *positional embeddings* (for position of tokens) and the *segment embeddings* (for modality type: vision or text).

### 3.4 Training with Sentence-Anatomy Dropout

During training, for each instance in each batch, we randomly drop a subset of anatomical tokens from the input and omit the corresponding sentences from the target radiology report; we term this training strategy **sentence-anatomy dropout**. In practice, for each training sample not all combinations of anatomical regions will be fit for dropout, since they must satisfy the following conditions:

1. Given a subset of anatomical tokens as input, the target output must be the full subset of sentences in the report that describe the corresponding anatomical regions;

2. Given a subset of sentences as the target output, anatomical tokens must be input for the full subset of described anatomical regions.

The above conditions are necessary since we reject the assumption that each sentence in the report describes only one anatomical region, as made by Tanida et al. (2023). We illustrate with examples of the different mappings in Table 1.

Let us consider a radiology report as a set of $L$ sentences $S = \{s_l\}_{l=1}^L$, each one describing the findings appearing in a different subset of anatomical regions $A_l \subseteq A$; and $P = \{\langle s_l, A_l \rangle\}_{l=1}^L$ the set of sentence-anatomy pairs of a report. To satisfy the two conditions above, we seek to discover the connected components in a graph where sentences are the nodes and an edge between two nodes represents an overlap of described anatomical regions between the two sentences. We describe in Appendix B the algorithm to identify the connected components for each CXR report and how we group the corresponding sentence-anatomy pairs to each connected component into $P_k \subseteq P$. We then define as $\mathcal{F} = \{P_k\}_{k=1}^K$ the *set of valid sentence-anatomy subsets* (see Appendix C for an example). During training, we randomly select one or more elements of $\mathcal{F}$ and then use the anatomical tokens as input and concatenate the corresponding sentences to create the target output.

## 4 Experiments

### 4.1 Datasets

We consider two open-source CXR imaging datasets: MIMIC-CXR (Johnson et al., 2019a,b;

Goldberger et al., 2000) and Chest ImaGenome (Wu et al., 2021; Goldberger et al., 2000). The MIMIC-CXR dataset comprises CXR image-report pairs. The Chest ImaGenome dataset includes additional annotations based on the MIMIC-CXR images and reports. In this paper, we train Faster R-CNN with the automatically extracted anatomical bounding box annotations from Chest ImaGenome, provided for 242,072 AnteroPosterior (AP) and PosteroAnterior (PA) CXR images. Chest ImaGenome also contains sentence-anatomy pairs annotations that we use to perform sentence-anatomy dropout. The longitudinal scans of each patient are obtained by ordering different studies based on the annotated timestamp and for each study, taking the most recent previous study as the prior. For this purpose, we only select AP or PA scans as priors (i.e. ignore lateral views). If multiple scans are present in a study, we consider the one with the highest number of non-zero anatomical tokens. In case of a tie, we select it randomly. In all experiments, we follow the train/validation/test split proposed in the Chest ImaGenome dataset.

### 4.2 Data pre-processing

We extract the *Findings* section of each report as the target text[4]. For the text input, we extract the *Indication field* from each report[5].

When training Faster R-CNN, CXRs are resized by matching the shorter dimension to 512 pixels (maintaining the original aspect ratio) and then cropped to a resolution of $512 \times 512$.

### 4.3 Metrics

We assess the quality of our model's predicted reports by computing three Natural Language Generation (NLG) metrics: BLEU (Papineni et al., 2002), ROUGE (Lin, 2004) and METEOR (Banerjee and Lavie, 2005). To better measure the clinical correctness of the generated reports, we also compute Clinical Efficiency (CE) metrics (Smit et al., 2020), derived by applying the CheXbert labeller to the ground truth and generated reports to extract 14 findings — and hence computing F1, precision and recall scores. In line with previous studies (Miura et al., 2021), this is computed by condensing the four classes extracted from CheXbert (positive, negative, uncertain, or no mention) into binary

---

[4] https://github.com/MIT-LCP/mimic-cxr/blob/master/txt/create_section_files.py
[5] https://github.com/jacenkow/mmbt/blob/main/tools/mimic_cxr_preprocess.py

classes of positive (`positive`, `uncertain`) versus negative (`negative`, `no mention`).

## 4.4 Implementation

We adopt the `torchvision` Faster R-CNN implementation, as proposed in Li et al. (2021). This consists of a ResNet-50 (He et al., 2016) and a Feature Pyramid Network (Lin et al., 2017) as the image encoder. We modify it and select the hyperparameters following Dalla Serra et al. (2023).

The two FC layers in the MLP projection layer have input and output feature dimensions equal to 2048 and include the bias term.

The encoder and the decoder of the Report Generator consist of 3 attention layers, each composed of 8 heads and 512 hidden units.

The MLP and the Report Generator are trained end-to-end for 100 epochs using a cross-entropy loss with Adam optimiser (Kingma and Ba, 2014) and the sentence-anatomy dropout training strategy. We set the initial learning rate to $5 \times 10^{-4}$ and reduce it every 10 epochs by a factor of 10. The best model is selected based on the highest F1-CE score.

We repeat each experiment 3 times using different random seeds, reporting the average in our results.

## 4.5 Baselines

We compare our method with previous SOTA works in CXR automated reporting, described in Section 2.1: R2Gen (Chen et al., 2020), R2GenCMN (Chen et al., 2021), $\mathcal{M}^2$ Transformer+fact$_{\text{ENTNLI}}$ (Miura et al., 2021), $A^{tok}$+TE+RG (Dalla Serra et al., 2023) and RGRG (Tanida et al., 2023). For all baselines, we keep the hyperparameters as the originally reported values. For a fair comparison, we use the same text and image pre-processing as proposed in this work and we re-train the baselines based on the Chest ImaGenome dataset splits.[6]

## 5 Results

### 5.1 Automated Reporting

Table 2 shows the effectiveness of the proposed method over the baselines, showing superior performance for most NLG and CE metrics. Compared to Dalla Serra et al. (2023), our method can achieve similar BLEU metrics and superior scores on the

remaining metrics, whilst providing also better controllability (and interpretability). Whilst both our method and that proposed by Tanida et al. (2023) tackle the controllability aspect, we show superior results in all metrics.

### 5.2 Ablation Study

We investigate the effect of incorporating prior CXR scans as input (**Priors**) and adopting sentence-anatomy dropout during training (**SA drop**).

**Full Reports** We evaluate the different configurations of our method using the full set of anatomical regions as input ($A_{target} = A$) and the full report as the target text. Table 3 shows the results; it can be seen that both adding priors and using sentence-anatomy dropout during training boost most metrics, with best overall performance obtained when combining the two mechanisms. This is illustrated qualitatively in Figure 3.

**Initial *vs* Follow-up Scans** We study the effect of the different components by dividing the test set into initial scans versus follow-up scans, resulting in 11,951 and 20,735 CXR report pairs respectively. The results are shown in Table 4. We note the improvement of our method over the baseline on both subsets, with the best results obtained when adding the priors alone or in combination with the sentence-anatomy dropout. It is worth noting that the benefit of including priors is also present for initial studies with no prior scans. We hypothesise that since the model can infer which are initial scans (using the fact that the prior anatomical tokens are all zero-vectors), it will correctly generate a more comprehensive report rather than focussing on progression or change of known findings.

**Partial Reports** To measure the controllability of our method, we evaluate the ability of the different configurations to generate partial reports given a subset of anatomical regions. For this purpose, we divide each report in the test set into its set of valid sentence-anatomy subsets $\mathcal{F}$ (Algorithm 1). We take the anatomical regions contained in each subset $\chi \subset \mathcal{F}$ as input and the corresponding sentences as the target output. We then obtain a total of 71,698 partial reports. The results in Table 5 show how adopting sentence-anatomy dropout enables a controllable method which correctly reports only on the anatomical regions presented as input and does not hallucinate the missing anatomical regions; this is further illustrated in Figure 4.

---

[6]We did not re-train Dalla Serra et al. (2023) and Tanida et al. (2023), as they already use that same dataset split.

| Method | NLG Metrics | | | | | | CE Metrics | | |
|---|---|---|---|---|---|---|---|---|---|
| | BL-1 | BL-2 | BL-3 | BL-4 | MTR | RG-L | F1 | P | R |
| R2Gen (Chen et al., 2020) | 0.381 | 0.248 | 0.174 | 0.130 | 0.152 | 0.314 | 0.431 | 0.511 | 0.395 |
| R2GenCMN (Chen et al., 2021) | 0.365 | 0.239 | 0.169 | 0.126 | 0.145 | 0.309 | 0.371 | 0.462 | 0.311 |
| $\mathcal{M}^2$ Tr. + fact$_{ENTNLI}$ (Miura et al., 2021) | 0.402 | 0.261 | 0.183 | 0.136 | 0.158 | 0.300 | 0.458 | 0.540 | 0.404 |
| $A^{tok}$ + TE + RG (Dalla Serra et al., 2023) | **0.490** | 0.363 | 0.288 | 0.237 | 0.213 | 0.406 | 0.537 | 0.585 | 0.496 |
| RGRG (Tanida et al., 2023) | 0.400 | 0.266 | 0.187 | 0.135 | 0.168 | - | 0.461 | 0.475 | 0.447 |
| *Ours* | 0.486 | **0.366** | **0.295** | **0.246** | **0.216** | **0.423** | **0.553** | **0.597** | **0.516** |

Table 2: Comparison of our proposed approach with previous methods. We show the **best results in bold**.

| Configuration | | NLG Metrics | | | | | | CE Metrics | | |
|---|---|---|---|---|---|---|---|---|---|---|
| Priors | SA drop | BL-1 | BL-2 | BL-3 | BL-4 | MTR | RG-L | F1 | P | R |
| - | - | 0.430 | 0.327 | 0.266 | 0.224 | 0.202 | 0.420 | 0.534 | 0.593 | 0.485 |
| ✓ | - | 0.456 | 0.347 | 0.283 | 0.239 | 0.210 | **0.428** | 0.548 | 0.577 | **0.522** |
| - | ✓ | 0.473 | 0.358 | 0.289 | 0.243 | 0.213 | 0.426 | 0.550 | **0.597** | 0.510 |
| ✓ | ✓ | **0.486** | **0.366** | **0.295** | **0.246** | **0.216** | 0.423 | **0.553** | **0.597** | 0.516 |

Table 3: Ablation study on incorporating prior CXR scans as input and adopting sentence-anatomy dropout during training. We report the NLG and CE results on the MIMIC-CXR test set.

| Configuration | | NLG Metrics | | | CE Metrics | | |
|---|---|---|---|---|---|---|---|
| Priors | SA drop | BL-4 | MTR | RG-L | F1 | P | R |
| | | **Initial Scans** | | | | | |
| - | - | 0.283 | 0.234 | 0.479 | 0.532 | 0.570 | 0.499 |
| ✓ | - | 0.303 | 0.244 | **0.490** | **0.543** | 0.563 | **0.524** |
| - | ✓ | 0.303 | 0.244 | 0.485 | 0.541 | 0.582 | 0.507 |
| ✓ | ✓ | **0.306** | **0.245** | 0.479 | 0.542 | **0.589** | 0.502 |
| | | **Follow-up Scans** | | | | | |
| - | - | 0.194 | 0.187 | 0.377 | 0.533 | 0.600 | 0.479 |
| ✓ | - | 0.206 | 0.194 | **0.383** | 0.550 | 0.583 | **0.521** |
| - | ✓ | 0.210 | 0.197 | 0.378 | 0.552 | **0.602** | 0.510 |
| ✓ | ✓ | **0.216** | **0.202** | 0.382 | **0.557** | 0.599 | 0.520 |

Table 4: NLG and CE results on the **Initial** and the **Follow-up** subsets of the MIMIC-CXR test set.

| Configuration | | NLG Metrics | | | CE Metrics | | |
|---|---|---|---|---|---|---|---|
| Priors | SA drop | BL-4 | MTR | RG-L | F1 | P | R |
| - | - | 0.113 | 0.187 | 0.291 | 0.549 | 0.519 | 0.583 |
| ✓ | - | 0.127 | 0.182 | 0.301 | 0.587 | 0.604 | 0.571 |
| - | ✓ | **0.225** | **0.226** | **0.467** | 0.667 | 0.651 | **0.683** |
| ✓ | ✓ | 0.223 | 0.225 | 0.462 | **0.672** | **0.663** | 0.680 |

Table 5: NLG and CE results computed on **partial reports** of the MIMIC-CXR test set, by dividing each report into its set of valid sentence-anatomy subsets.

To further assess the quality of our method and each of its components, we measure the length distribution of the predicted reports, similar to Chen et al. (2020), showing that our method more closely matches the ground truth distribution than baseline methods; see results in Appendix D.

## 6 Limitations

While the proposed method shows state-of-the-art results on CXR automated reporting, end-to-end report generation from CXR images requires further research to reach the clinical accuracy needed to be useful as a diagnostic tool. We note that our eval-

uation is itself limited since the CheXbert labeller reportedly has an accuracy of only 0.798 F1 (Smit et al., 2020) and thus we do not expect to measure perfect scores on our clinical efficiency metrics.

Our method focuses only on CXR and adapting it to other types of medical scans might be challenging. First, due to the 2D nature of CXRs compared to other types of 3D scans (*e.g.*, CT, MRI). Second, we strongly rely on the Chest ImaGenome dataset and its annotations. These are automatically extracted and similar sentence-anatomy annotations could be extracted for radiology reports from other types of scans. However, as the same authors pointed out, there are some known limitations of their NLP and the region extraction pipelines; for instance, clinical findings may not be properly extracted from a follow-up report which may be as simple as "No change is seen". Hence, some refinement of the pipelines with or without additional manual input might be required.

## 7 Conclusion

This work focussed on two key aspects of CXR automated reporting: *controllability* and *longitudinal CXR representation*. We proposed a simple yet effective solution to align, concatenate and fuse the anatomical representations of two subsequent CXR scans into a joint representation used as the visual input to a language model for automated reporting. We then proposed a novel training strategy termed *sentence-anatomy dropout*, to supervise the model to link each anatomical region to the corresponding output sentences. This gives the user more control and easier interpretability of the model predictions.

| Ground Truth | Baseline | Ours | *Comments* |
|---|---|---|---|
| There are persisting small to moderate bilateral pleural effusions with subjacent atelectasis as well as pulmonary vascular engorgement and mild interstitial septal thickening. The right infrahilar opacity is unchanged. There is enlargement of the cardiac silhouette unchanged. Calcification of the aortic arch is present. | Moderate to severe cardiomegaly is stable. Moderate bilateral pleural effusions are unchanged. There is mild pulmonary edema. There is no focal consolidation or pneumothorax. | Moderate to large right and small left pleural effusions are unchanged. There is mild pulmonary vascular congestion and interstitial edema. There is a moderate right pleural effusion with associated right lower lobe atelectasis. Moderate cardiomegaly is stable. The aortic knob is calcified. No pneumothorax. No acute osseous abnormalities. | *The report predicted by our method shows a more accurate description of the findings and higher recall compared to the baseline.* |
| In comparison to the prior chest radiograph all lines and tubes have been removed. The bilateral lung aeration has improved dramatically. There is a small left pleural effusion. There is a subtle right basilar opacity. Heart size is normal. The mediastinal and hilar contours are normal. The pulmonary vasculature is normal. No pneumothorax is seen. | Compared to the prior study there is no significant interval change. | Interval removal of the endotracheal tube. The feeding tube has been removed. The right internal jugular catheter has been removed. The cardiomediastinal and hilar contours are within normal limits. The aorta is tortuous. There is a small left pleural effusion with adjacent atelectasis. There is no pneumothorax. | *The report predicted by our method correctly describes the removal of the different tubes.* |
| Stable chronically elevated right hemidiaphragm with moderate right lower lobe atelectasis. Small stable breast calcification projects adjacent to the left ventricle as is seen on ct. No pneumothorax or pleural effusion. Left lung clear. Heart size mediastinal contour and hila appear normal. No bony abnormality. | No focal consolidation pleural effusion pneumothorax or pulmonary edema is seen. Heart size is normal. Mediastinal contours are normal. | Elevation of the right hemidiaphragm is unchanged from prior. The lungs are clear without focal consolidation. No pleural effusion or pneumothorax is seen. Heart and mediastinal contours are within normal limits with tortuous aortic contour. There is elevation of the right hemidiaphragm. | *Our proposed method shows an improvement compared to the baseline, but there is still missing information such as a comparison with a previous CT scan.* |

Figure 3: Qualitative results of full reports generation. We compare the reports generated by the baseline (without adding prior scans and sentence-anatomy training) and the proposed solution with the ground truth. We highlight using different colours the segments of the reports that are commented on in the right column.

| Anat. Regions | Ground Truth | Baseline | Ours |
|---|---|---|---|
| spine | Moderate anterior osteophytes along the lower thoracic spine appear unchanged. | **The heart is at the upper limits of normal size. The aortic arch is calcified. The mediastinal and hilar contours appear unchanged. There is no pleural effusion or pneumothorax. The lungs appear clear.** | Mild degenerative changes are similar along the thoracic spine. |
| cardiac silhouette | Moderate to severe enlargement of the cardiac silhouette is unchanged. | Moderate enlargement of the cardiac silhouette is re-demonstrated. **Mediastinal and hilar contours are unchanged. Pulmonary vasculature is not engorged. No focal consolidation pleural effusion or pneumothorax is present. There are no acute osseous abnormalities**. | Moderate to severe cardiomegaly is re-demonstrated. |
| abdomen, cardiac silhouette, mediastinum, upper mediastinum | The nasogastric tube has been removed. The heart and mediastinum are within normal limits. | The heart size is normal. The **hilar** and mediastinal contours are normal. **There has been interval resolution of the previously seen small left-sided pneumothorax. There has been slight interval improvement of the previously seen small right pleural effusion. There is no evidence of focal consolidations concerning for pneumonia.** | The nasogastric tube has been removed. The cardiomediastinal silhouette is unremarkable. |
| left lung, left lower lung zone left costophrenic angle, left hilar structures, right lung, right lower lung zone, right costophrenic angle, right hilar structures | There are areas of streaky atelectasis at the bilateral lung bases. There are persistent prominent interstitial markings which suggest chronic interstitial abnormality versus mild interstitial edema. The lungs remain hyperinflated. There is no pleural effusion or pneumothorax. No focal consolidation is seen. | Lung volumes are low. Linear opacities in the bilateral lower lungs are most consistent with subsegmental atelectasis. There is no focal consolidation pleural effusion or pneumothorax. **The cardiomediastinal silhouette is unchanged.** | There is pulmonary vascular congestion and mild interstitial pulmonary edema. Linear bibasilar opacities are most consistent with atelectasis. There is no pleural effusion or pneumothorax. |

Figure 4: Qualitative results of partial reports generation. From left to right: the subset of anatomical regions $A_{target}$ we want to report, the ground truth partial reports, the reports generated by the baseline (without adding prior scans and sentence-anatomy training) based on $A_{target}$ and those generated by our proposed method. We indicate in **red** the hallucination on the missing anatomical regions.

We showed the effectiveness of the proposed solution on the MIMIC-CXR dataset where it gives state-of-the-art results. Moreover, we evaluated through extensive ablations how the different components help to generate better reports in different setups: *full report generation*, *partial report generation*, and *Initial vs. Follow-up report generation*.

In future, this method could be integrated with more advanced language models such as Touvron et al. (2023) or OpenAI (2023), or alternative techniques such as Dalla Serra et al. (2023). Further, when considering the patient history, the prior CXR scan might usefully be augmented with other types of imaging and associated radiology reports, clinical notes, clinical letters, and lab results. In future work, we will look to extend our method to a broader multimodal approach considering more data inputs.

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

# A  Anatomical Regions & Findings

Below we show the lists of 36 anatomical regions and 71 clinical findings used in this paper.

| Anatomical Regions | | | |
|---|---|---|---|
| abdomen | left clavicle | mediastinum | right lower lung zone |
| aortic arch | left costophrenic angle | right apical zone | right lung |
| cardiac silhouette | left hemidiaphragm | right atrium | right mid lung zone |
| carina | left hilar structures | right cardiac silhouette | right upper abdomen |
| cavoatrial junction | left lower lung zone | right cardiophrenic angle | right upper lung zone |
| descending aorta | left lung | right clavicle | spine |
| left apical zone | left mid lung zone | right costophrenic angle | svc |
| left cardiac silhouette | left upper abdomen | right hemidiaphragm | trachea |
| left cardiophrenic angle | left upper lung zone | right hilar structures | upper mediastinum |

| Findings | | | |
|---|---|---|---|
| airspace opacity | cyst/bullae | linear/patchy atelectasis | pneumothorax |
| alveolar hemorrhage | diaphragmatic eventration (benign) | lobar/segmental collapse | prosthetic valve |
| aortic graft/repair | elevated hemidiaphragm | low lung volumes | pulmonary edema/hazy opacity |
| artifact | endotracheal tube | lung cancer | rotated |
| aspiration | enlarged cardiac silhouette | lung lesion | scoliosis |
| atelectasis | enlarged hilum | lung opacity | shoulder osteoarthritis |
| bone lesion | enteric tube | mass/nodule (not otherwise specified) | spinal degenerative changes |
| breast/nipple shadows | fluid overload/heart failure | mediastinal displacement | spinal fracture |
| bronchiectasis | goiter | mediastinal drain | sub-diaphragmatic air |
| cabg grafts | granulomatous disease | mediastinal widening | subclavian line |
| calcified nodule | hernia | multiple masses/nodules | superior mediastinal mass/enlargement |
| cardiac pacer and wires | hydropneumothorax | pericardial effusion | swan-ganz catheter |
| chest port | hyperaeration | picc | tortuous aorta |
| chest tube | ij line | pigtail catheter | tracheostomy tube |
| clavicle fracture | increased reticular markings/ild pattern | pleural effusion | vascular calcification |
| consolidation | infiltration | pleural/parenchymal scarring | vascular congestion |
| copd/emphysema | interstitial lung disease | pneumomediastinum | vascular redistribution |
| costophrenic angle blunting | intra-aortic balloon pump | pneumonia | |

Table 6: Complete set of 36 anatomical regions and 71 findings used to supervise the anatomy localisation and the finding detection tasks, as annotated in the Chest ImaGenome dataset (https://physionet.org/content/chest-imagenome/1.0.0/).

## B  Algorithm for discovering valid sentence-anatomy subsets

---

**Algorithm 1** Find set of valid sentence-anatomy subsets.

---

**Input:** set of $\langle$sentence, regions$\rangle$ pairs from a single CXR report, $P$
**Output:** set of valid sentence-anatomy subsets, $\mathcal{F}$

1: **function** FINDVALIDSUBSETS($P$)
2:     $\mathcal{F} \leftarrow$ empty set
3:     $P_i \leftarrow$ set populated with the first $\langle$sentence, regions$\rangle$ pair in $P$
4:     $P_{remaining} \leftarrow$ set of $\langle$sentence, regions$\rangle$ pairs in $P$ not assigned to $P_i$
5:     $R(P_i) \leftarrow$ regions in $P_i$
6:     $R(P_{remaining}) \leftarrow$ regions in $P_{remaining}$
7:     **while** $P_{remaining} \neq \{\}$ **do**
8:         **while** $R(P_i) \cap R(P_{remaining}) \neq \{\}$ **do**
9:             **for** $\langle$sentence, regions$\rangle \in P_{remaining}$ **do**
10:                 **if** regions $\cap R(P_i) \neq \{\}$ **then**
11:                     $P_i \leftarrow$ include $\langle$sentence, regions$\rangle$
12:                 **end if**
13:             **end for**
14:             $P_{remaining} \leftarrow$ set of $\langle$sentence, regions$\rangle$ pairs in $P$ not assigned to $P_i$ nor any $P_k \in \mathcal{F}$
15:             $R(P_i) \leftarrow$ regions in $P_i$
16:             $R(P_{remaining}) \leftarrow$ regions in $P_{remaining}$
17:         **end while**
18:         $\mathcal{F} \leftarrow$ include $P_i$
19:         $P_i \leftarrow$ set populated with the first $\langle$sentence, regions$\rangle$ pair in $P_{remaining}$
20:     **end while**
21:     **return** $\mathcal{F}$
22: **end function**

---

# C  Example of Report as Sentence-Anatomy Pairs

**GT Report**

*'The lungs are hyperinflated with flattening of the diaphragms suggestive of underlying COPD. The heart is mildly enlarged. The aorta is tortuous and diffusely calcified. Mediastinal and hilar contours otherwise are unremarkable. Pulmonary vascularity is not engorged. No focal consolidation, pleural effusion or pneumothorax is identified. There are minimal streaky bibasilar atelectatic changes. No acute osseous abnormalities are present. Mild multilevel degenerative changes are seen in the thoracic spine.'*

**Chest ImaGenome sentence-anatomies pairs:**

$$P = \{\langle s_l, A_l \rangle\}_{l=1}^{L}$$

| Sentence | Anatomical Regions |
|---|---|
| 'The lungs are hyperinflated with flattening of the diaphragms suggestive of underlying COPD.' | ['right lung', 'left lung', 'right hemidiaphragm', 'left hemidiaphragm'] |
| 'Pulmonary vascularity is not engorged.' | ['right lung', 'right hilar structures', 'left lung', 'left hilar structures'] |
| 'No focal consolidation, pleural effusion or pneumothorax is identified.' | ['right lung', 'right costophrenic angle', 'left lung', 'left costophrenic angle'] |
| 'There are minimal streaky bibasilar atelectatic changes.' | ['right lung', 'right lower lung zone', 'left lung', 'left lower lung zone'] |
| 'Mediastinal and hilar contours otherwise are unremarkable.' | ['right hilar structures', 'left hilar structures', 'mediastinum', 'upper mediastinum'] |
| 'The aorta is tortuous and diffusely calcified.' | ['mediastinum', 'aortic arch'] |
| 'The heart is mildly enlarged.' | ['cardiac silhouette'] |
| 'No acute osseous abnormalities are present.' | ['right clavicle', 'left clavicle', 'spine'], |
| 'Mild multilevel degenerative changes are seen in the thoracic spine.' | ['spine'] |

**Set of valid sentence-anatomy subsets:**

$$\mathcal{F} = \{P_k : P_k \subseteq P\}_{k=1}^{K}$$

| Sentences | Anatomical Regions |
|---|---|
| 'The lungs are hyperinflated with flattening of the diaphragms suggestive of underlying COPD.', | ['right lung', 'left lung', 'right hemidiaphragm', 'left hemidiaphragm'] |
| 'Pulmonary vascularity is not engorged.', | ['right lung', 'right hilar structures', 'left lung', 'left hilar structures'] |
| 'No focal consolidation, pleural effusion or pneumothorax is identified.', | ['right lung', 'right costophrenic angle', 'left lung', 'left costophrenic angle'] |
| 'There are minimal streaky bibasilar atelectatic changes.', | ['right lung', 'right lower lung zone', 'left lung', 'left lower lung zone'] |
| 'Mediastinal and hilar contours otherwise are unremarkable.', | ['right hilar structures', 'left hilar structures', 'mediastinum', 'upper mediastinum'] |
| 'The aorta is tortuous and diffusely calcified.' | ['mediastinum', 'aortic arch'] |
| 'No acute osseous abnormalities are present.', | ['right clavicle', 'left clavicle', 'spine'] |
| 'Mild multilevel degenerative changes are seen in the thoracic spine.' | ['spine'] |
| 'The heart is mildly enlarged.' | ['cardiac silhouette'] |

Figure 5: Example of sentence-anatomy annotations of a report and its set of valid sentence-anatomy subsets.

# D Report Length

The length of a report corresponds to the number of words contained. We compute this for the full reports generated from the full set of anatomical regions and for the partial reports derived from the set of valid sentence-anatomy subsets. In Figure 6 (left) we see that the distribution of the proposed method is closer to the distribution of the GT reports. In Figure 6 (right), we note that adopting the sentence-region dropout strategy allows the method to generate partial reports with a length distribution closer to the GT partial reports. These results provide further evidence of the improvement of the proposed method over the baseline (without adding prior scans and sentence-anatomy training).

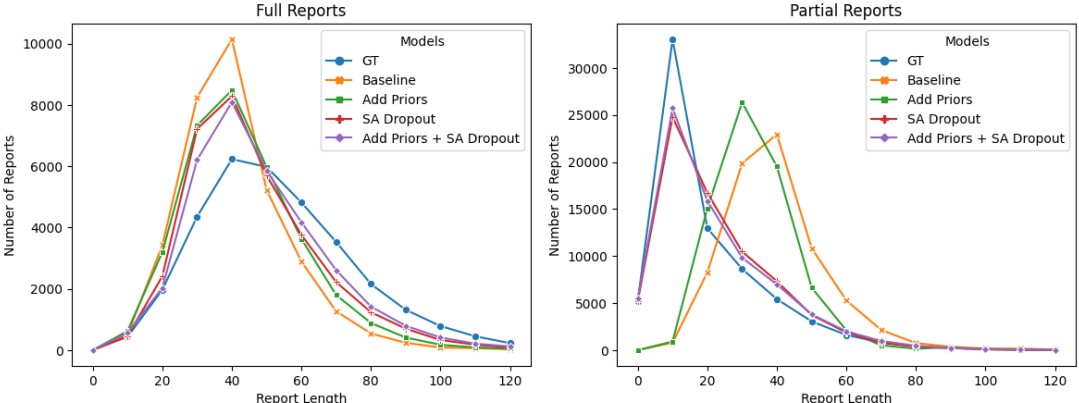

Figure 6: Length distribution of the predicted reports compared to the GT reports. The length of a report corresponds to the number of words.