# OpenReview forum: "Controllable Chest X-Ray Report Generation from Longitudinal Representations"
_EMNLP/2023/Conference — EMNLP 2023 Findings_

### Official Review · Reviewer_1BQa · 2023-07-22

**Soundness:** 3

**Excitement:**

3: Ambivalent: It has merits (e.g., it reports state-of-the-art results, the idea is nice), but there are key weaknesses (e.g., it describes incremental work), and it can significantly benefit from another round of revision. However, I won't object to accepting it if my co-reviewers champion it.

**Paper Topic And Main Contributions:**

The paper focuses on addressing the challenges in automating radiology reporting, which typically involves detailed text descriptions of medical scans and their clinical findings. The key problem the paper tackles is the need for high accuracy and interpretability in clinical deployment of automated reporting systems.

The main contributions of this paper can be summarized as follows:

1. Longitudinal Representation Learning: The paper introduces a novel approach for incorporating prior scan information into the reporting process. Unlike previous methods that do not use the prior study as input, this work proposes longitudinal representation learning. It aligns, concatenates, and fuses the current and prior visual information to create a joint longitudinal representation.
2. Sentence-Anatomy Dropout: This paper introduces a training strategy for controllability in report generation called sentence-anatomy dropout.

**Questions For The Authors:**

Question A: The paper seems to be an extension of the paper (Anonymous,  2023) 'Task-aware anatomical tokens for chest x-ray automated reporting'. Based on the comparison in Table 2 and Table 3, I notice that most of the improvement seems to come from the inherited method from the paper. In some aspects, the proposed method seems to be worser than (Anonymous,  2023). If the authors wrote the two papers, why not combine the methods and propose that as a single paper. In addition, it is unclear to me what are the major differences between this two papers and what makes the most significant contributions to performance difference.

Question B: Section 3.4 mentions the differences between the current paper and RGPG (Tanida et al.,2023). Table 2 and 3 shows that even without  Priors and SA drop, it is better than RGPG. Thus, I wonder with the same underlying method, what is the difference between base+SA drop and base+RGPG.

Question C: Line 341 mentions that you cropped the CXR. Are the cropped-out regions mostly background or they contain valid information relating to reporting?

Question D: Line 373 you used 100 epochs  for training. Does it converge? Does it need that many epochs to converge?

Question E: Line 381, you mention that you select models  based on the best score. I am not sure what models did you selected from? Did you selected the best model among different random  seeds?


**Reasons To Accept:**

This paper combines and extends the effective strategy from existing works and demonstrate improved performance on the task of automatic radiology reporting. The major contributions are:

1. Approach for Radiology Reporting: The paper introduces a novel approach for automating radiology reporting by leveraging anatomical representations of prior scans. This shift from traditional image-level features to anatomical representations presents a new perspective and can potentially enhance the accuracy and clinical relevance of automated reports.

2.Sentence-Anatomy Dropout for Controllability: The introduction of sentence-anatomy dropout as a training strategy allows the report generator model to predict sentences corresponding to specific anatomical regions. This fine-grained controllability in report generation is crucial for clinicians who require targeted and interpretable reports, and it addresses a significant limitation of previous automated reporting systems.

**Reasons To Reject:**

1. Lack of Clarity on Major Contributions: The paper appears to combine ideas from existing works rather than introducing a novel structure. It is unclear whether the reported performance improvement is solely attributable to this paper or results from the combination of ideas from the previous work (Anonymous, 2023). This ambiguity makes it challenging to ascertain the specific contributions of this paper and may raise concerns about the novelty and originality of the approach.

2. Limited Performance Improvement: While the paper seems to extend the prior work (Anonymous, 2023) and introduce the methods of Longitudinal Representation Learning and Training with Sentence-Anatomy Dropout, the reported performance improvement in table 2 and 3 are not substantial when compared to (Anonymous, 2023). This raises questions about the significance and effectiveness of the proposed extensions.

3. Lack of Sufficient Novelty in Methodologies: The paper introduces Longitudinal Representation Learning as an extension of Karwande et al. (2022) by using Faster R-CNN (Ren et al., 2015). This approach does not seem to offer a significant advancement beyond the existing methods, potentially limiting its contribution to the NLP community.

4. Some of the experiments can be better designed. For example, Figure 4 compare results from with and without adding prior scans and sentence-anatomy training). For sure that if the baseline never sees anat. region-sentence pairs in training, then it will not generate sentence properly if asking for region descriptions. This experiment relating to hallucination can be better designed.

**Reproducibility:**

5: Could easily reproduce the results.

**Reviewer Confidence:**

4: Quite sure. I tried to check the important points carefully. It's unlikely, though conceivable, that I missed something that should affect my ratings.

**Typos Grammar Style And Presentation Improvements:**

Unclear writing which  may be improved:
LINE 018: 'an existing visual input format of anatomical tokens,'
LINE 054: 'monitoring' --> 'disease monitoring'
LINE 136:  'by ...'

---

> ### Author Rebuttal · Authors · 2023-08-28
>
> We thank the reviewer for their valuable feedback and their proposed improvements to make the writing clearer. We are also encouraged that they found the technical novelty compelling.
>
> We begin by addressing the concern raised in the “Reasons to Reject” section regarding Longitudinal Representation Learning; we believe the other concerns are addressed below in response to specific questions. We have indeed taken a similar approach to Karwande et al. (2022), albeit for the target task of text generation and with a Transformer architecture as the workhorse model. Our novelty lies in proposing a region-level mechanism for the use of prior images as context for text report generation (to the best of our knowledge, we are the first to do so). Since we are inputting the visual tokens to a large Transformer model (and training end-to-end), for practical reasons we chose to use a more computationally efficient method than Karwande et al, employing an MLP rather than GCN, and performing *unified* region detection and encoding in Faster R-CNN (instead of cropping the RoIs from the CXR and encoding them separately).
>
> Please find our answers to the questions raised below.
>
> **Question A: The paper seems to be an extension of the paper (Anonymous, 2023) 'Task-aware anatomical tokens for chest x-ray automated reporting'. Based on the comparison in Table 2 and Table 3, I notice that most of the improvement seems to come from the inherited method from the paper. In some aspects, the proposed method seems to be worser than (Anonymous, 2023). If the authors wrote the two papers, why not combine the methods and propose that as a single paper. In addition, it is unclear to me what are the major differences between this two papers and what makes the most significant contributions to performance difference.**
>
> - This paper proposes methodological improvements (sentence-anatomy dropout training strategy, incorporation of prior scans to enable longitudinal representations) that improve performance compared to (Anonymous, 2023), in particular on the clinical efficiency (CE) metrics which we believe to be the most important and which we use for best parameter model selection. We agree that the NLG metrics are sometimes very close between the 2 methods, however, we consider these must be treated with caution since they measure linguistic rather than semantic similarity to the ground truth reports, and therefore can be contradictory e.g. a generated sentence *“There is severe pulmonary vascular congestion”* vs ground truth of *“There is no pulmonary vascular congestion”* would score highly on NLG metrics but not on CE metrics. Compared to (Anonymous, 2023) our method has the additional disadvantage that it does not use the two-step architecture used by (Anonymous, 2023); this explains why the ablation results in Table 3 are worse than the inherited method. We intend that the two-step architecture could also be used with our method to maximise performance; we plan to do this investigation as future work (as we state in our Conclusion). We further note that our sentence-anatomy dropout training strategy confers controllability, which was not an explicit feature of (Anonymous, 2023) although an emergent learned relationship between tokens and output was observed in some instances - see Figure 2 in (Anonymous, 2023) - and served as motivation for this paper.
>
> **Question B: Section 3.4 mentions the differences between the current paper and RGPG (Tanida et al.,2023). Table 2 and 3 shows that even without Priors and SA drop, it is better than RGPG. Thus, I wonder with the same underlying method, what is the difference between base+SA drop and base+RGPG.**
>
> - Indeed, our method without Priors and SA drop shows better results than RGRG, which can likely be attributed to the broader context offered by providing the model with the full set of anatomical regions when generating the report as opposed to generating a sentence at the time from a single anatomical region. Since the design of RGRG is centered around how to generate a suitable sentence from a given (single) anatomical region (e.g. region selection module), it is difficult to see which elements of the base method are compatible and could be meaningfully combined whilst retaining the contributions of RGRG. However, we did not put extensive effort into attempting to combine the two since Tanida et al. is a recently released paper (17th April) which we became aware of shortly before the submission deadline of June 23rd.
>
> **Question C: Line 341 mentions that you cropped the CXR. Are the cropped-out regions mostly background or they contain valid information relating to reporting?**
>
> - The regions that are cropped out typically consist of the upper and lower sections of the CXRs, as CXRs usually have a vertical orientation. These sections usually comprise the neck and the lower abdominal area of the patient, depending on how the patient is positioned i.e. not the chest region. Consequently, the relevant regions are still present after cropping the image (e.g. lungs, heart, etc.).  However, we agree there is a risk in rare cases that such image pre-processing cuts out some relevant peripheral regions.
>
> **Question D: Line 373 you used 100 epochs for training. Does it converge? Does it need that many epochs to converge?**
>
> - Yes, the model tends to converge around 70 epochs. Conservatively, we train for a further 30 epochs to ensure the model has trained. However, we select the best model from an earlier epoch using the best F1 score on the validation data.
>
> **Question E: Line 381, you mention that you select models based on the best score. I am not sure what models did you selected from? Did you selected the best model among different random seeds?**
>
> - No, as we mention above, here we refer to the process of choosing the best model for 1 run (i.e. at which epoch the training converged). Apologies, we can see that the sentence ordering for Lines 378-381 in Section 4.4 is confusing (the sentence about best model selection should be moved to the previous paragraph), we will clarify this in the camera-ready paper.

---

### Official Review · Reviewer_pwef · 2023-07-25

**Typos Grammar Style And Presentation Improvements:** N/A
**Soundness:** 3

**Excitement:**

4: Strong: This paper deepens the understanding of some phenomenon or lowers the barriers to an existing research direction.

**Missing References:**

N/A

**Paper Topic And Main Contributions:**

This paper demonstrates longitudinal representation learning on chest X-ray scans by fusing learned feature representations from both a current and a prior image, and further introduces sentence-anatomy dropout for controlling the text report output, by limiting the report output to a subset of anatomical (SA) regions provided as input. Improved performance is claimed on MIMIC-CXR-based datasets.


**Questions For The Authors:**

A. In Section 3.2, it is stated that there is one token for each of the N anatomical regions, and that the longitudinal representation is obtained by concatenating the respective anatomical tokens, for the two CXRs. It might be clarified as to the outcome were the anatomical region not to be detected, in one or both CXRs.

B. In Section 5.1, For the baseline results as reported in Table 2, it might be clarified as to whether the comparison methods are able to exploit longitudinal data (i.e. additional prior visit images), since it appears that prior data was indeed used with the proposed approach, from the results in Table 3.

C. Moreover, SA dropping is also exploited in achieving the results reported for the proposed method in Table 2. It might be clarified as to how the relevant SA to keep/drop were chosen, for the Chest ImaGenome dataset - were the bounding box annotations used, and if so, were these annotations expected to correspond to the expected output in the radiology text report? In any case, this would seem to attribute part of the superiority of the proposed method, to the usage of additional information not available to comparison methods.

D. In Section 5.2, hallucination on missing anatomical regions is mentioned. Does this imply that the actual scan(s) do not actually contain the anatomical regions reported on? In any case, this might be illustrated with the corresponding images if possible in an expanded Figure 4, for better understanding.


**Reasons To Accept:**

 - Comprehensive ablation experiments on respective contribution of prior data, as well as SA dropping.


**Reasons To Reject:**

 - Unclear if the comparisons against existing methods are entirely fair, given possible use of prior and SA-based inputs not available to such methods


**Reproducibility:**

4: Could mostly reproduce the results, but there may be some variation because of sample variance or minor variations in their interpretation of the protocol or method.

**Reviewer Confidence:**

4: Quite sure. I tried to check the important points carefully. It's unlikely, though conceivable, that I missed something that should affect my ratings.

---

> ### Author Rebuttal · Authors · 2023-08-28
>
> We thank the reviewer for their valuable feedback. We are encouraged that they found the ablation experiments comprehensive to support the two main novel aspects of this paper: longitudinal scan integration and SA dropout.
>
> In response to the comment in the "Reasons To Reject" section (*“unclear if the comparisons against existing methods are entirely fair, given possible use of prior and SA-based inputs not available to such methods”*), we agree that the proposed mechanisms could improve existing methods and hope this paper provides evidence to try introducing our ideas and finding the best marriage with other existing strategies. The intention behind our current results in Table 2 is to illustrate how the two principal novel contributions proposed within this paper when integrated into a standard Transformer architecture, yield SOTA outcomes without employing any of the alternative solutions featured in existing methods (architectural design, reinforcement learning, and knowledge grounding, among others).
>
> Please find our answers to the questions raised by the reviewer below.
>
> **A. In Section 3.2, it is stated that there is one token for each of the N anatomical regions, and that the longitudinal representation is obtained by concatenating the respective anatomical tokens, for the two CXRs. It might be clarified as to the outcome were the anatomical region not to be detected, in one or both CXRs.**
>
> - As stated in Section 3.1 (lines 226-228), when an anatomical region is not detected, the corresponding token embedding is set to a zero vector representation. This is equivalent to the step of removing an anatomical region that is used in our proposed sentence-anatomy dropout strategy. Therefore, we expect the model to omit reporting on any undetected regions.
>
> **B. In Section 5.1, For the baseline results as reported in Table 2, it might be clarified as to whether the comparison methods are able to exploit longitudinal data (i.e. additional prior visit images), since it appears that prior data was indeed used with the proposed approach, from the results in Table 3.**
>
> - Indeed, the prior CXR is used in the results presented in Table 2. This component is a fundamental aspect of our approach and forms the “longitudinal” aspect of our method. Existing methods have largely ignored prior data, and we consider that it is beyond the scope of this paper to attempt to find the best way to incorporate prior data into each existing method, by making ad-hoc adjustments that potentially deviate far from the original methodologies as conceived by their respective authors. However, we hope that our work showing one method of introducing prior data and its impact on performance might encourage other authors to similarly incorporate prior data into their methods.
>
> **C. Moreover, SA dropping is also exploited in achieving the results reported for the proposed method in Table 2. It might be clarified as to how the relevant SA to keep/drop were chosen, for the Chest ImaGenome dataset - were the bounding box annotations used, and if so, were these annotations expected to correspond to the expected output in the radiology text report? In any case, this would seem to attribute part of the superiority of the proposed method, to the usage of additional information not available to comparison methods.**
>
> - As stated in Section 3.4 (lines 287-306), performing SA dropout requires generating the set of valid sentence-anatomy subsets for each report. This is achieved by identifying *"the connected components in a graph where the sentences are the nodes and an edge between two nodes represents an overlap of described anatomical regions between the two sentences"*. This step can be straightforwardly performed using the detailed annotations from the Chest ImaGenome dataset, in which each sentence is annotated with the mentioned anatomical regions. During training, this is used to randomly dropout a subset of sentences from the target output (contained in one or more connected components) and the corresponding anatomical regions from the input. Comparatively, the sole other method within the existing literature utilizing the Chest ImaGenome dataset's sentence-anatomy annotations is RGRG (Tanida et al., 2023), which generates sentences focused on individual regions sequentially. In Section 3.4 and Table 1 (lines 272-286), we explain our rationale for rejecting this approach and instead advocating for SA dropout. The results of RGRG (Tanida et al., 2023) are also presented in Table 2 for comparison. We agree that including sentence/anatomy annotations using the method proposed in this paper or alternative supervision strategies, could improve other existing methods, and hope this paper provides evidence to try this.
>
> **D. In Section 5.2, hallucination on missing anatomical regions is mentioned. Does this imply that the actual scan(s) do not actually contain the anatomical regions reported on? In any case, this might be illustrated with the corresponding images if possible in an expanded Figure 4, for better understanding.**
>
> - No, we do not consider the case that the actual scans may not contain some anatomical regions (although this may occasionally be the case and we would hope that the relevant regions are simply not detected if so). In Section 5.2 and Figure 4, our objective is to illustrate the impact of the SA dropout training strategy by showing that at inference time we get different effects when we block the visual information from selected regions.  Namely, we select a subset of regions that we wish to report on by zeroing out the vector representations of the "missing" anatomical regions (effectively simulating their non-detection); therefore, no direct visual information is passed to the reporting model about these regions. When the SA dropout training strategy is not used, descriptions are still generated for these "missing" regions. We will augment Figure 4 with CXR images, highlighting the selected regions, provided that we obtain the necessary permissions from the MIMIC-CXR authors.

---

### Official Review · Reviewer_kJZG · 2023-08-05

**Soundness:** 5

**Excitement:**

5: Transformative: This paper is likely to change its subfield or computational linguistics broadly. It should be considered for a best paper award. This paper changes the current understanding of some phenomenon, shows a widely held practice to be erroneous in someway, enables a promising direction of research for a (broad or narrow) topic, or creates an exciting new technique.

**Paper Topic And Main Contributions:**

This paper proposes a method to incorporate a previous image with a current image to do radiology report generation. A significant challenges is aligning the previous and current images so that comparisons can be made and this paper proposes using an R-CNN, trained for anatomy localization, to do this. The paper then inputs image tokens and the indication for the current scan into a few transformer layers for predicting the radiology report.

**Questions For The Authors:**

Is there a way to combine this approach with an image-based pretraining strategy beyond multi-label prediction to address the long tail of rare diseases?

Did the authors consider trying an LLM instead of the few transformer layers that were used?

**Reasons To Accept:**

This paper presents a straightforward method to make use of prior images for radiology report generation. The authors present compelling results on public datasets, outperforming other recent baselines.

**Reasons To Reject:**

As the image encoder was trained for multi-label prediction, it may not represent the long tail of rare diseases as well. This is in contrast to a visual-language pretrained image encoder which may encode more rare disease information.

**Reproducibility:**

4: Could mostly reproduce the results, but there may be some variation because of sample variance or minor variations in their interpretation of the protocol or method.

**Reviewer Confidence:**

4: Quite sure. I tried to check the important points carefully. It's unlikely, though conceivable, that I missed something that should affect my ratings.

---

> ### Author Rebuttal · Authors · 2023-08-28
>
> We thank the reviewer for their positive feedback. We are encouraged they found the paper interesting and the results compelling. Please find our answers to the reviewer’s questions below.
>
> **Is there a way to combine this approach with an image-based pretraining strategy beyond multi-label prediction to address the long tail of rare diseases?**
> - Apart from the use of Faster R-CNN we have only explored using a ResNet-101 (pretrained on ImageNet or the 71 clinical findings) integrated with the Transformer encoder-decoder, similar to what most of the existing automated reporting methods have adopted. However, we are aware of the challenge posed by addressing the long tail of rarer diseases and we thank the reviewer for the insightful suggestion of using a vision-language pretrained image encoder.
>
> **Did the authors consider trying an LLM instead of the few transformer layers that were used?**
> - As we are limited on computational resources (i.e. all our models were trained on a single Nvidia RTX A5000 GPU), we did not try using an LLM or increasing the number of parameters of the Transformer. Also, in the first place, we wanted to keep the setup simple and focused on our proposed methodology. We agree with the reviewer that trying larger LLMs (e.g. GPT, LLaMa) is a good idea and (as we state in our Conclusion) we plan to do this investigation as future work.

---

### Meta-Review · Area_Chair_8Sxm · 2023-09-16

**Recommendation:** 3

**Metareview:**

The authors describe an interesting and well-executed study. The reviewers agree that the quality of the experiments is high and report no issues with clarity. The main concerns relate to the originality of the approach compared to previous work by the same authors, as well as the significance of the results compared to past experiments and baselines. The author response does a good job alleviating some of these concerns and showcasing the novelty and value of the proposed approach. In my view, the main question at this point is whether this paper should be published in its current form or if there should be a bigger paper that combines this approach with the one proposed in prior work and provides comparison to enhanced baselines.

Quality:

Pros:

-	“This paper presents a straightforward method to make use of prior images for radiology report generation. The authors present compelling results on public datasets, outperforming other recent baselines.”

-	“Comprehensive ablation experiments on respective contribution of prior data, as well as SA dropping”

Cons:

-	“As the image encoder was trained for multi-label prediction, it may not represent the long tail of rare diseases as well.”

Originality:

Pros:

-	“This shift from traditional image-level features to anatomical representations presents a new perspective and can potentially enhance the accuracy and clinical relevance of automated reports.”

-	“The introduction of sentence-anatomy dropout as a training strategy allows the report generator model to predict sentences corresponding to specific anatomical regions. This fine-grained controllability in report generation is crucial for clinicians who require targeted and interpretable reports, and it addresses a significant limitation of previous automated reporting systems.”

Cons:

-	“The paper appears to combine ideas from existing works rather than introducing a novel structure. It is unclear whether the reported performance improvement is solely attributable to this paper or results from the combination of ideas from the previous work (Anonymous, 2023). This ambiguity makes it challenging to ascertain the specific contributions of this paper and may raise concerns about the novelty and originality of the approach.” In their response, the authors highlight that there are methodological changes that lead to improvements on the most significant metrics, as well as to some disadvantages shown in the ablation study. I agree with the point raised by the reviewer that a combination between the two studies into one paper will likely provide better value to the community.

-	“The paper introduces Longitudinal Representation Learning as an extension of Karwande et al. (2022) by using Faster R-CNN (Ren et al., 2015). This approach does not seem to offer a significant advancement beyond the existing methods, potentially limiting its contribution to the NLP community.” The authors respond that: “Our novelty lies in proposing a region-level mechanism for the use of prior images as context for text report generation (to the best of our knowledge, we are the first to do so). Since we are inputting the visual tokens to a large Transformer model (and training end-to-end), for practical reasons we chose to use a more computationally efficient method than Karwande et al, employing an MLP rather than GCN, and performing unified region detection and encoding in Faster R-CNN (instead of cropping the RoIs from the CXR and encoding them separately).” Based on my understanding as a metareviewer, the contribution here is mainly in the computational efficiency of the new approach, which I believe is an important advancement.

Significance:

The main issue here relates to the fact that the comparison to some baselines may not be fair since they were not provided the additional data that the main method utilizes.

-	“Unclear if the comparisons against existing methods are entirely fair, given possible use of prior and SA-based inputs not available to such methods”. Another reviewer notes: “For sure that if the baseline never sees anat. region-sentence pairs in training, then it will not generate sentence properly if asking for region descriptions. This experiment relating to hallucination can be better designed.” The authors respond that: “We agree that including sentence/anatomy annotations using the method proposed in this paper or alternative supervision strategies, could improve other existing methods, and hope this paper provides evidence to try this.” The question about fairness of comparison remains – why not compare to these previous methods by providing them with the additional data as well?

Clarity:

No major concerns reported in the reviews.

---

### Decision · Program_Chairs · 2023-10-07

**Decision:**

Accept-Findings

**Comment:**

The authors describe an interesting and well-executed study. The reviewers agree that the quality of the experiments is high and report no issues with clarity. The main concerns relate to the originality of the approach compared to previous work by the same authors, as well as the significance of the results compared to past experiments and baselines. The author response does a good job alleviating some of these concerns and showcasing the novelty and value of the proposed approach. In my view, the main question at this point is whether this paper should be published in its current form or if there should be a bigger paper that combines this approach with the one proposed in prior work and provides comparison to enhanced baselines.

Quality:

Pros:

-	“This paper presents a straightforward method to make use of prior images for radiology report generation. The authors present compelling results on public datasets, outperforming other recent baselines.”

-	“Comprehensive ablation experiments on respective contribution of prior data, as well as SA dropping”

Cons:

-	“As the image encoder was trained for multi-label prediction, it may not represent the long tail of rare diseases as well.”

Originality:

Pros:

-	“This shift from traditional image-level features to anatomical representations presents a new perspective and can potentially enhance the accuracy and clinical relevance of automated reports.”

-	“The introduction of sentence-anatomy dropout as a training strategy allows the report generator model to predict sentences corresponding to specific anatomical regions. This fine-grained controllability in report generation is crucial for clinicians who require targeted and interpretable reports, and it addresses a significant limitation of previous automated reporting systems.”

Cons:

-	“The paper appears to combine ideas from existing works rather than introducing a novel structure. It is unclear whether the reported performance improvement is solely attributable to this paper or results from the combination of ideas from the previous work (Anonymous, 2023). This ambiguity makes it challenging to ascertain the specific contributions of this paper and may raise concerns about the novelty and originality of the approach.” In their response, the authors highlight that there are methodological changes that lead to improvements on the most significant metrics, as well as to some disadvantages shown in the ablation study. I agree with the point raised by the reviewer that a combination between the two studies into one paper will likely provide better value to the community.

-	“The paper introduces Longitudinal Representation Learning as an extension of Karwande et al. (2022) by using Faster R-CNN (Ren et al., 2015). This approach does not seem to offer a significant advancement beyond the existing methods, potentially limiting its contribution to the NLP community.” The authors respond that: “Our novelty lies in proposing a region-level mechanism for the use of prior images as context for text report generation (to the best of our knowledge, we are the first to do so). Since we are inputting the visual tokens to a large Transformer model (and training end-to-end), for practical reasons we chose to use a more computationally efficient method than Karwande et al, employing an MLP rather than GCN, and performing unified region detection and encoding in Faster R-CNN (instead of cropping the RoIs from the CXR and encoding them separately).” Based on my understanding as a metareviewer, the contribution here is mainly in the computational efficiency of the new approach, which I believe is an important advancement.

Significance:

The main issue here relates to the fact that the comparison to some baselines may not be fair since they were not provided the additional data that the main method utilizes.

-	“Unclear if the comparisons against existing methods are entirely fair, given possible use of prior and SA-based inputs not available to such methods”. Another reviewer notes: “For sure that if the baseline never sees anat. region-sentence pairs in training, then it will not generate sentence properly if asking for region descriptions. This experiment relating to hallucination can be better designed.” The authors respond that: “We agree that including sentence/anatomy annotations using the method proposed in this paper or alternative supervision strategies, could improve other existing methods, and hope this paper provides evidence to try this.” The question about fairness of comparison remains – why not compare to these previous methods by providing them with the additional data as well?

Clarity:

No major concerns reported in the reviews.